# Fertility Intention to Have a Third Child in China following the Three-Child Policy: A Cross-Sectional Study

**DOI:** 10.3390/ijerph192215412

**Published:** 2022-11-21

**Authors:** Ni Ning, Jingfei Tang, Yizhou Huang, Xiangmin Tan, Qian Lin, Mei Sun

**Affiliations:** 1Xiangya School of Nursing, Central South University, Changsha 410013, China; 2College of Engineering and Design, Hunan Normal University, Changsha 410081, China; 3Xiangya School of Public Health, Central South University, Changsha 410078, China

**Keywords:** fertility intention, third child, family policy, cross-sectional study

## Abstract

China’s three-child policy was implemented in May 2021 to stimulate a rise in fertility levels. However, few previous studies have explored fertility intentions to have a third child and have only focused on childless or one-child populations, resulting in a gap in findings between fertility intention and fertility behavior. Thus, we conducted a nationwide cross-sectional study on 1308 participants with two children. Results showed that only 9.6% of participants reported planning to have a third child and 80.2% of the population had heard of the policy but had no idea of the detailed contents. Participants with two daughters (OR = 3.722, 95% CI = 2.304–6.013) were willing to have one more child. Instrumental values (OR = 1.184, 95% CI = 1.108–1.265) and policy support (OR = 1.190, 95% CI = 1.124–1.259) were the facilitators. Perceived risk (OR = 0.883, 95% CI = 0.839–0.930) and higher educational level (OR = 0.693, 95% CI = 0.533–0.900) were the leading barriers to having one more child. Therefore, the government should deepen parents’ understanding of the “three-child policy” and devise ways of reducing the negative impacts of having a third child to boost the intention to have more children. Our proposed approach can also be used to better understand the reasons for low fertility rates in other countries.

## 1. Introduction

In recent years, the declining fertility rate has been a real problem in China and has become a global phenomenon [1,2]. According to documents published by the United Nations, fertility rates in many middle- and high-income countries have fallen below-replacement fertility levels (under 2.1 births per woman) by 2020 [3]. Fertility decline is the main driver behind the rapid aging of the global population [4]. Sustained below-replacement fertility produces a population with relatively fewer young people and more old people [5]. As the population ages, this may lead to labor shortages and challenges for the healthcare system, hindering economic and social development [6].

In 2000, China entered a phase of persistently low fertility levels [7]. The latest China Census data showed a fourth consecutive decline in the birth rate from 2017 to 2021 [8]. In early 2022, the National Bureau of Statistics released data showing that the number of newborns in 2021 was 10.62 million, a decrease of 1.38 million compared to 2020 [9]. In response to the highly complex and uncertain fertility environment, China has adopted a gradual strategy of fertility policy adjustment, ranging from the abolition of birth spacing restrictions to the implementation of a separate two-child policy to a comprehensive two-child policy, meeting the diverse fertility needs of different populations [10]. However, China’s fertility rate continued to decline, falling into what international academics classify as a very low fertility level by 2021 [7]. To avoid the rapid aging of the population because of the continuous decline in the fertility rate, China started to implement the three-child policy in 2021. A range of early childhood education services, maternity leave, maternal and child health services and a maternity insurance system has also been introduced to increase the willingness of Chinese couples to have a third child [11].

The success of the fertility policy depends firstly on how a couple’s intentions to have a child can be realized [12]. Fertility intention can be defined as a psychological mechanism that expresses fertility desires based on individual or family preferences for children [13,14]. Investigating couples’ fertility intentions and related factors is essential to inform the fertility policy for its successful implementation.

However, fertility intentions are only a tendency and do not necessarily translate into fertility behavior [14]. The gap between intentions and behaviors has become a regular phenomenon in developed and developing countries [15]. Compared with measuring fertility intentions among the unmarried or childless population, the specific fertility plans of married couples, especially those who have already given birth, are more relevant to reality, considering policy, family and personal conditions [16]. Previous studies which investigated fertility intention and its associated factors often recruited from all of the reproductive age population, including unmarried and childless populations [10]. In contrast with previous research, we recruited couples who already have two children, because their intentions for a third child are closer to realistic behavior, in order to minimize the gap between intentions and behaviors, eliminate the influence of other factors and obtain a clear picture of the attitudes of people of reproductive age toward child-bearing.

Research on fertility intentions has become a central focus of discussion on the low fertility rate issue [17]. Focusing on the fertility intentions of the child-bearing age population, especially the willingness to have more children, can inform policies to promote fertility in middle- and high-income countries, which are consistently below the fertility replacement rate. The purpose of this study is to assess the willingness to have a third child among the general population of child-bearing age who already have two children in China after the announcement of the three-child policy. It also examines relevant factors affecting the willingness to have a child, providing evidence for China to address population issues and improve fertility support policies.

## 2. Materials and Methods

### 2.1. Study Design and Sample

A national cross-sectional survey was conducted online in China on 2 October 2021 using a convenience sampling method. To contact the potential participants, we used WeChat, the main social media platform in China with 1.27 billion active users [18], to publish the recruitment announcements and weblink for the electronic questionnaire. The inclusion criteria were as follows: (1) have two children in the family, (2) married women and men, (3) age ranges of 20–49 years old and (4) consent to participate. Participants who were pregnant with a third child or had undergone permanent sterilization were not eligible for inclusion. This study followed the Strengthening the Reporting of Observational Studies in Epidemiology (STROBE) reporting guideline.

### 2.2. Measurements and Instruments

The theory of planned behavior, the basic theoretical model of intention and behavior, has significance in predicting fertility behavior. As the theory noted [19], fertility intentions are usually measured as the timing and number of childbearing episodes. Furthermore, intention to have or not have a child can be explained by three determinants: attitudes, subjective norm and perceived control. In addition, individuals’ background factors, including personal, social and informational factors, in turn influence these three immediate determinants of fertility intentions (see Figure 1). Based on the theory of planned behavior and previous studies, a questionnaire was prepared to collect information combining the following five main components:

(1) Related sociodemographic characteristics: Gender, age, residence (city or rural), living area (classified into eastern, central and western China), education level, employment status, house ownership and household income (one person’s income per month) are included. According to the data from China’s Bureau of Statistics, household income (one person per month) is divided into three levels: low-income level (<5000 RMB), middle-income level (5000–10,000 RMB) and high-income level (>10,000 RMB).

(2) Measurement of fertility intention:

Fertility intention of a third birth: As the dependent variable and primary outcome, this was measured by one question: “Do you intend to have another child?” with the response options being: “do not intend, intend” accordingly.

Desired number of children: Participants were asked to indicate how many children they want to have in the ideal case with the following options: 0 children, 1 child, 2 children, ≥3 children and does not matter.

Desired inter-pregnancy interval (IPI): Participants were asked, “How many years after the upcoming/recent birth would you like to have the next child?”. Besides a space to specify the number of years, women could choose the options “I will not want additional children” or “I do not know.”

(3) Knowledge of the three-child policy: Participants were asked, “How much do you know about the three-child policy?” with the following response options: never heard, heard about it, know a little and very clear.

(4) Attitude toward the three-child policy: Participants were asked, “What’s your attitude towards the three-child policy?” with the following response options: fully accepted, reluctantly accepted, unacceptable, unreality should change, and have no idea.

(5) Determinants of fertility intentions scale: Zhou [20] created the scale to explore the factors affecting fertility intention according to the three determinants of planned behavior theory. This multidimensional scale is rated on the following six-point scale: 0 = absolutely untrue, 1 = mostly untrue, 2 = somewhat untrue, 3 = somewhat true, 4 = mostly true and 5 = absolutely true. Twenty-four items assess six dimensions: instrumental values, emotional values, social support, policy support, perceived risk and available resources. The fertility intention scale has good reliability and validity and the internal consistency reliability of each dimension is 0.688–0.768. The internal consistency reliability of each dimension was 0.79–0.91 in this study.

### 2.3. Data Collection

An online survey platform, Wenjuanxing: www.wjx.cn (accessed on 28 October 2021), was used to generate a weblink for the electronic questionnaire. The participants were informed that they were taking part in the study voluntarily and that the data would be anonymously processed without identifying their personal information when they opened the weblink. Participants had two options to select after reading the informed consent. If they selected the option “refused to take part in this study,” the webpage would remind participants to choose the reason for refusing and then automatically close the questionnaire. Next, the webpage showed five questions (e.g., “How many children do you have?”) relevant to our inclusion and exclusion criteria. If participants selected an option that did not meet the criteria, the webpage would close the questionnaire automatically.

All survey items needed to be answered before the questionnaire could be submitted to ensure the effectiveness of data collection. Participants were assigned identification numbers, each of which could be used only once. After data collection, the questionnaire responses were collated and examined by research team members.

### 2.4. Data Analysis

All data analyses were conducted with SPSS 26.0. Outcomes were defined as the fertility intention to have a third child and divided into two groups, “have intention” and “have no intention.” The normality of the data was tested using the Kolmogorov-Smirnov test. For categorical variables, we calculated frequencies and proportions. For continuous variables, media and interquartile range (IQR) were calculated as data were not normally distributed. A subgroup analysis according to gender was performed in this study. The difference in the sociodemographic characteristics and the score of factors influencing fertility intention scale for females and males were examined using the Pearson χ^2^ test and Kruskal-Wallis test. A multivariate logistic regression model was used to test the potential factors influencing the intention to have a third child. The binary dependent variable was the fertility intentions to have a third child (no or yes). Independent variables with *p* < 0.05 were added to the multivariate logistic regression model and the backward regression method was applied to select variables for the final model. The odds ratio (OR), 95% confidence interval (95% CI) and *p*-value for each independent variable were calculated. Statistical significance was considered at *p* < 0.05. Graphs were prepared in Graphpad Prism (version 8.0.1).

### 2.5. Ethical Considerations

All participants recruited in this study had agreed to the informed consent. Participants were also informed that they could withdraw from the study at any time and could contact the investigators if they had any questions about the questionnaire. The responses of any participant in this study were confidential and only accessible to the research team. The authors declared that all the experiments of this study complied with the current laws of China in which they were performed. The study was approved by the Ethics Committee of Behavioral and Nursing research in the School of Nursing of Central South University Ethical Committee (No. 2021145).

## 3. Results

### 3.1. Response Rate and Study Population

Of the 1392 returned questionnaires, six individuals refused to participate because of fear of information disclosure and 78 were excluded because of incomplete questionnaires. Finally, a total of 1308 respondents (887, 67.8 female, 421, 32.2% male), were enrolled in the final analysis, with an effective response rate of 94%.

The sociodemographic characteristics of all participants are summarized in Table 1. The majority (65%) of participants were 30–39 years old and 78.4% lived in central China. The number of people living in urban areas is approximately three times that in rural areas. Approximately 84.4% of the participants had a college degree or higher educational background. A large percentage of participants (93.4%) were employed and 80.0% had a moderate or above household income. The proportion of those with a son and a daughter was the highest, accounting for 56.2% of the total participants.

### 3.2. Measurement of Fertility Intention

Among all the participants with two children, although 150 (11.5%) desired to have more than or equal to three children in the ideal case, only 125 (9.6%) intended to have a third child in reality. Most (74%) reported that the desired family size was two children per family. More results about the fertility intention of different groups can be found in Table 2.

### 3.3. Knowledge of the Three-Child Policy

Figure 2 shows that 80.2% of the population have heard of the policy but had no idea of the detailed contents overall. Among those who intended to have a third child, most (45.6%) know part of the contents and almost a fifth of the participants (20%) said they were clear about it. Nearly half of those who did not intend to have a third child (47.5%) had only heard of the policy and those who knew the policy well accounted for just 14%.

### 3.4. Attitude toward the Three-Child Policy

When asked about their opinion toward the three-child policy, 62.6% of the respondents reported that they accepted the policy, whereas 27.3% thought the policy was unrealistic and should be changed. Among those who intended to have a third child, the majority fully accepted (62.4%) and reluctantly accepted (23.2%) this policy. Among those who did not want to have a third child, most also fully accepted (32.1%) and reluctantly accepted (28.1%) this policy. Approximately 29.2% of participants thought it was unrealistic, which is three times more than those who intended to have a third child (see Figure 3).

### 3.5. Factors Influencing Fertility Intention

Intention to have a third child was statistically significantly different in terms of gender (χ^2^ = 17.477, *p* = 0.000), education level (χ^2^ = 21.690, *p* = 0.000), employment status (χ^2^ = 7.435, *p* = 0.024) and gender of two children (χ^2^ = 33.690, *p* = 0.000), but no significant differences were found in fertility intentions by age (χ^2^ = 4.279, *p* = 0.233), income (χ^2^ = 1.988, *p* = 0.370) and house ownership (χ^2^ = 2.744, *p* = 0.433).

Table 3 shows the scores of factors affecting the fertility intention scale. The results showed that fertility intention was significantly associated with six dimensions: instrumental values (χ^2^ = 54.487, *p* = 0.000), emotional values (χ^2^ = 5.769, *p* = 0.000), social support (χ^2^ = 10.690, *p* = 0.001), family policy support (χ^2^ = 18.797, *p* = 0.000), perceived risk (χ^2^ = 59.060, *p* = 0.000) and available resources (χ^2^ = 7.821, *p* = 0.005).

In Table 4, we conducted a logistic regression analysis to determine the factors influencing the intention to have a third child. The willingness to have a third child decreased significantly as education level increases (OR = 0.693, 95% CI = 0.533–0.900). People with two daughters (OR = 3.722, 95% CI = 2.304–6.013) are more likely to have another child than those with one son and one daughter. Instrumental values (OR = 1.184, 95% CI = 1.108–1.265) and policy support (OR = 1.190, 95% CI = 1.124–1.259) positively affect the intention to have a third child. Perceived risk (OR = 0.883, 95% CI = 0.839–0.930) had a negative impact on the intention to have a third child.

## 4. Discussion

A decline in fertility level has been recognized as an issue of global health concern and is most prominent in China. This study used a population-based survey to investigate fertility intention to have a third child among married couples in China with two children. It also examined the factors influencing intention to access useful information for policymakers about the short-term effects of the new three-child policy. Overall, participants had a low level of knowledge of the three-child policy and an extremely low intention regarding a third birth. Notably, those families with two daughters were more willing to have one more child. Instrumental values and policy support were the facilitators and perceived risk and higher educational level were the leading barriers to having one more child.

The proportion of participants with the intention of a third birth was 9.6% in our sample, which was lower than a national survey conducted by Yan [10], reporting that 12.2% of 15,332 participants would consider having a third child. This difference may be because of the different sample populations. Yan’s survey recruited childbearing-age participants no matter how many children they had or even if they had no child, which may confound the measurement of the intention to have a third child. Zhu, et al. [21] reported that 9.4% of reproductive-age couples who have two children in Shanghai intended to have a third child, which was rather similar to the values reported in the present survey. A previous study indicated that women who had previously given birth were likely to report a lower desired number of children compared with those women who had never given birth [22]. Furthermore, according to behavioral control in Ajzen’s theory, the facilitates and barriers towards one’s fertility behavior are not fully perceived and under control when formulating one’s fertility intention, which then causes an intention–behavior gap [23]. For instance, as the number of children grows, the burden of caregiving increases, which could turn a previously positive intention into a negative one late [12].

To the best of our knowledge, this study is the first to assess the knowledge and attitude of the three-child policy among the child-bearing age population. Our results indicated that most of the reproductive-age population with two children, the target population of the three-child policy, had a low level of policy knowledge. Furthermore, participants who intended to have a third child showed a higher level of policy knowledge than those who did not intend to have a third child, which is consistent with the results of Gong and Wang’s study [24] on the association of family policy awareness and marital intentions in Japan. The failure of a family policy may be due to the policy itself but could also be attributed to low policy awareness and perceived accessibility among the target population [24]. Thus, in addition to focusing on policy availability, policymakers should prioritize public awareness of the family policy. Surprisingly, we found that more than 30% of participants with no intention to have a third child fully agreed and accepted the three-child policy. Conducting a qualitative study in future research to explore this unexpected phenomenon more deeply would provide further insight.

We found that participants with two daughters were willing to have a third child, with a preference for sons. The reasons are analyzed below. First, by tradition, China practiced a strict patrilineality, patriarchy and patrilocality system. Men were dominant in wealth inheritance, living arrangements, family line continuity and intrahousehold power structure and women attached themselves to their husbands [25]. Compared with daughters who eventually marry out, sons could economically provide farm laborers in the agricultural society, provide social security support to aged parents, culturally carry on the family line and enhance the family status in the community [26,27]. According to Confucianism, of the three most unfilial practices, having no male offspring is the biggest. The preference for sons has been prevalent throughout Chinese history. Second, the cost of marriage for daughters was generally lower than that for sons in China because the bridegroom in China should provide the bride price, the wedding ceremony and the wedding apartment [28]. Thus, individuals with two daughters do not yet have these predictable financial burdens and are, therefore, more intended to have at least one son in the strong son preference context.

An interesting finding was that the most important factors influencing a couple’s intention to have a third child in present-day China are related to the instrumental values of children, compared with available resources, which was the mainly focus of previous studies [29,30]. We consider the following reasons in order to explain this phenomenon in our study. First, along with the general economic development in the past three decades in China, systematic improvements in access to health care and individual income level have contributed to the improvements in available resources for maternal and child health [31]. Second, items with the highest score on the instrumental values dimension were the benefits to the child’s mental health and family stability. An only child may feel lonely while growing up. Siblings can provide more opportunities for social interaction and mutual learning, which may benefit their mental health [32]. Many studies have established parenthood as a protective factor against separation or divorce [33], which is important in the Chinese familism culture. In China, familism not only prioritizes family, but also advocates the continuity of the family line as a life mission. In other words, fertility benefits family stability.

Perceived risk, including fertility risk and child-bearing burden, are the leading barriers to having one more child. A literature review indicated health concerns about fertility risk significantly affect couples’ intentions to have more children [34]. Kjerulff, et al. [35] also found that women who had a caesarean delivery were less likely to have two or three children than those who had a vaginal delivery because most couples consider multiple caesarean deliveries a high-risk. Reducing the percentage of caesarean deliveries and improving the management of fertility risk prevention is necessary for increasing fertility intention to have more children. Furthermore, most couples often feel tired of caring for the children while balancing this with their work. Studies found that three-generation co-residence positively impacted fertility intention because the intergenerational transmission of child-bearing behavior can help couples balance childcare and work [36]. A policy encouraging three-generation living (either co-residence or close proximity) was used in Singapore [37], is perhaps appropriate in China or other Asian countries which have similar cultural backgrounds and should be considered by the government.

Consistent with other research, education level is one of the most established socioeconomic determinants of fertility intentions and participants with higher level education were less likely to have a third child [10,38,39]. The previous study indicated that the third birth intention of those with bachelor’s and master’s degrees decreased by 37.7% and 44.6%, compared with participants with high school degrees [10]. This phenomenon could be explained by the fact that adult children who serve as their primary caregivers provide economic support for their elderly parents through the traditional Chinese culture. However, elderly individuals with a higher level education in China can live on their pensions; thus, the motivation for having more children declines [32]. More importantly, women with higher education are no longer satisfied with being full-time housewives and are mainly focused on pursuing their career development [40]. These women do not intend to have more children in order to balance work and family relationships.

We hope to explore the factors affecting the intention to have a third child after implementing a new family policy by using this nationwide survey in China, which is an appropriate research setting and holds significant implications for other East Asian countries. China, Korea and Japan face common problems of sustained below-replacement fertility and share traditional gender roles and family norms [41]. Thus, our analysis of factors impacting the intention to have more children in China could facilitate a better understanding of how to improve family policies promoting fertility intentions in other East Asian societies.

This study has several limitations. First, this study’s results were based on WeChat users and the number of participants was not balanced across regions or gender, which has certain deviations from the national average. In future investigations, multi-center and gender-matched research focusing on people of different economic and cultural backgrounds should be conducted to avoid this bias. Second, this cross-sectional survey was based on self-reported questionnaires at a one-time point, which failed to draw a causal relationship; we need to design longitudinal studies to examine reproductive behaviors over the years to draw more accurate conclusions. Finally, this study only focused on the impact of six dimensions in determinants of fertility intention scale. Even if the items in this scale could reveal fertility risk to some extent, the mode of delivery and obstetric history were not included in this study, which needs to be explored.

## 5. Conclusions

This study gives an overview of the intention to have one more child in China after implementing the new three-child policy. The results showed that the intention to have a third child and the knowledge level of the three-child policy are extremely low. Parents’ understanding of the new family policy must be deepened and supporting measures to create a favorable fertility context must be implemented. Limited by the cross-sectional study design, we need to develop more qualitative studies to explore barriers among populations who do not intend have a third child.

## Figures and Tables

**Figure 1 ijerph-19-15412-f001:**
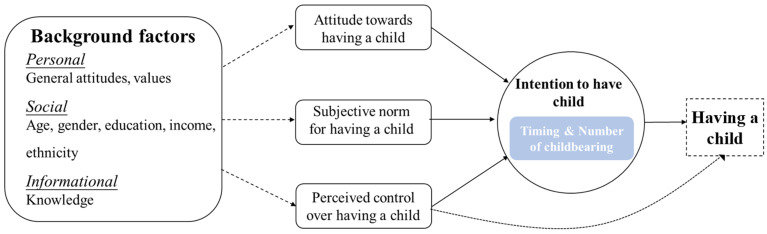
Theory of planned behavior applied to fertility intentions.

**Figure 2 ijerph-19-15412-f002:**
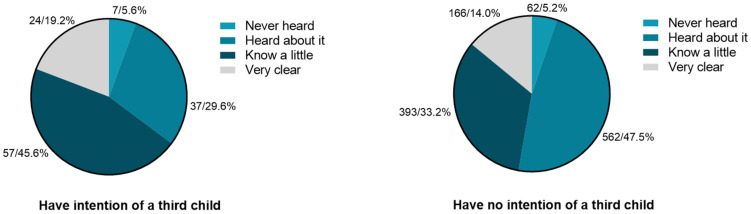
Public knowledge of the three-child policy.

**Figure 3 ijerph-19-15412-f003:**
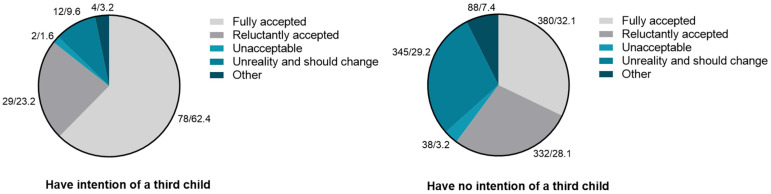
Public attitude toward the three-child policy.

**Table 1 ijerph-19-15412-t001:** Characteristics of all participants (N = 1308).

		Intention to Have a Third Child			
Characteristic	Total n (%)	Yes (N = 125) n (%)	No (N = 1183) n (%)	X^2^/H	*p* Value
Gender				17.477	0.000 *
Male	421 (32.2)	61 (48.8)	360 (30.4)		
Female	887 (67.8)	64 (51.2)	823 (68.6)		
Age				4.279	0.233
≤29	121 (9.3)	17 (13.6)	104 (8.8)		
30–34	393 (30.0)	31 (24.8)	362 (30.6)		
35–39	457 (34.9)	46 (36.8)	411 (34.7)		
≥40	337 (25.8)	31 (24.8)	306 (25.9)		
Living area				2.659	0.265
East China	207 (15.8)	26 (20.8)	181 (15.3)		
Central China	1026 (78.4)	93 (74.4)	933 (78.9)		
West China	75 (5.8)	6 (4.8)	69 (5.8)		
Residence				0.197	0.657
City	984 (75.2)	92 (73.6)	892 (75.4)		
Rural	324 (24.8)	33 (26.4)	291 (24.6)		
Education level				21.690	0.000 *
High school or below	204 (15.6)	37 (29.6)	167 (14.1)		
College	762 (58.3)	56 (44.8)	706 (59.7)		
Master or above	342 (26.1)	32 (25.6)	342 (26.2)		
Employment status				7.435	0.024 *
Employed	1035 (79.1)	90 (72.0)	945 (79.9)		
Self-employed	187 (14.3)	28 (22.4)	159 (13.4)		
Unemployed	86 (6.6)	7 (5.6)	79 (6.7)		
Income (one person per month)				1.988	0.370
<5000	262 (20.0)	31 (24.8)	231 (19.5)		
5000–10,000	548 (41.9)	50 (40.0)	498 (42.1)		
>10,000	498 (38.1)	44 (35.2)	454 (38.4)		
House ownership				2.744	0.433
Tenancy	135 (10.3)	18 (14.4)	117 (9.9)		
Housing with loan	696 (53.2)	61 (48.8)	635 (53.7)		
Housing without loan	355 (27.1)	34 (27.2)	321 (27.1)		
Parents’ house	122 (9.3)	12 (9.6)	110 (9.3)		
Gender of two children				33.690	0.000 *
2 males	285 (21.8)	18 (14.4)	267 (22.6)		
2 females	288 (22.0)	53 (42.4)	235 (19.9)		
1 male and 1 female	735 (56.2)	54 (43.2)	681 (57.6)		

Note: * *p* < 0.05.

**Table 2 ijerph-19-15412-t002:** Measurement of fertility intention (N = 1308).

Dimensions	Male (N = 421) n/%	Female (N = 887) n/%
Fertility intention of a third birth		
Have intention	61 (14.5)	64 (7.2)
Have no intention	360 (85.5)	823 (92.8)
Desired number of children		
0	5 (1.2)	6 (0.7)
1	35 (8.3)	68 (7.7)
2	294 (69.8)	674 (76.0)
≥3	60 (14.3)	90 (10.1)
Does not matter	27 (6.4)	49 (5.5)
Desired intra-pregnancy interval		
Want the third child within 1–3 years	35 (57.4)	38 (59.4)
Want the third child within 4 years or more	11 (18.0)	10 (15.6)
Unsure	15 (24.6)	16 (25.0)
Have no intention	360 (85.5)	823 (92.8)

**Table 3 ijerph-19-15412-t003:** Scores of determinants of fertility intentions scale (N = 1183).

Dimensions	Items	M (IQR)		H	*p*
		Intended	Unintended		
		(N = 125)	(N = 1183)		
Instrumental values		14.00 (5.5)	10.00 (7)	68.752	0.000 *
	Fertility can ease pension pressure.	3.00 (2)	2.00 (2)		
	More children are good for children’s mental health.	4.00 (2)	3.00 (2)		
	Fertility is beneficial to family stability.	4.00 (2)	3.00 (2)		
	Fertility can increase motivation at work.	3.00 (3)	2.00 (2)		
Emotional values		17.00 (6)	16.00 (7)	5.769	0.016 *
	I love children.	5.00 (1)	4.00 (2)		
	Children are my spiritual comfort.	4.00 (2)	4.00 (2)		
	Children can enrich my life experience.	4.00 (2)	4.00 (2)		
	I feel happy with my children.	5.00 (1)	4.00 (2)		
Social support		13.00 (6)	11.00 (5)	35.841	0.000 *
	My family members want to have more children.	3.00 (2)	2.00 (3)		
	My family atmosphere is harmonious and suitable for children’s development.	4.00 (2)	4.00 (2)		
	My family members will help me when I have trouble parenting.	4.00 (3)	3.00 (3)		
	My friends will help me when I meet trouble parenting.	3.00 (3)	2.00 (3)		
Policy support		14.00 (5.5)	9.00 (7)	113.384	0.000 *
	The fertility policy will affect my intention to have children.	4.00 (2)	2.00 (3)		
	Knowledge of fertility policies has a strong influence on fertility intentions.	4.00 (2)	3.00 (3)		
	The government can promote the implementation of a three-child policy with enough support measures.	3.00 (2)	2.00 (3)		
	The three-child policy has boosted my fertility intention.	3.00 (3)	1.00 (2)		
Perceived risk		14.00 (6)	16.00 (7)	29.536	0.000 *
	Fertility risk has significant impact on fertility intention.	4.00 (2.5)	5.00 (2)		
	Various occupations have a significant impact on fertility intention.	3.00 (3)	4.00 (3)		
	Working hours have a significant impact on fertility intention.	4.00 (2)	5.00 (2)		
	Raising children will add to my childcare burden.	3.00 (2)	5.00 (2)		
Available resource		14.00 (4)	13.00 (5)	7.821	0.005 *
	Domestic early childhood education and care can meet my needs.	3.00 (2)	2.00 (2)		
	The domestic medical level can meet my needs.	4.00 (2)	4.00 (2)		
	The domestic postnatal services can meet my needs.	4.00 (1)	3.00 (2)		
	My income level is sufficient for me to raise my children.	5.00 (2)	5.00 (0)		

Note: * *p* < 0.05. Abbreviations: M, median; IQR, interquartile range.

**Table 4 ijerph-19-15412-t004:** Logistic regression analysis for factors impacting intention to have a third child.

Variables	Beta Coefficient		*p*	OR	95%CI
	*B*	St. Error			
Gender of two children					
1 boy 1 girl	Reference				
2 boys	−0.031	0.306	0.919	0.969	[0.532, 1.764]
2 girls	1.314	0.245	0.000	3.722	[2.304, 6.013]
Fertility decisions maker					
Couples together	Reference				
Myself	−0.632	0.307	0.039	0.531	[0.291, 0.970]
Spouse	0.230	0.380	0.546	1.258	[0.598, 2.649]
Education level	−0.367	0.134	0.006	0.693	[0.533, 0.900]
Instrumental values	0.169	0.034	0.000	1.184	[1.108, 1.265]
Emotional values	−0.066	0.035	0.060	0.937	[0.875, 1.003]
Family policy support	0.174	0.029	0.000	1.190	[1.124, 1.259]
Perceived risk	−0.124	0.026	0.000	0.883	[0.839, 0.930]

Note: Gender, education level, employment status and gender of two children were included in the logistic regression model as covariates along with the six dimensions of determinants of fertility intention scale. Abbreviations: CI, confidence interval; OR, odds ratio.

## Data Availability

The data presented in this study are available on request from the corresponding author. The data are not publicly available due to reasons of privacy.

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
