# Peer review of "Fertility Intention to Have a Third Child in China following the Three-Child Policy: A Cross-Sectional Study"

_ijerph, 2022, doi:10.3390/ijerph192215412_

Round 1

Reviewer 1 Report

Thank you for performing this very useful piece of work. I support your approach of targeting couples with already 2 children to target how they would respond to the new policy.

I would like to see more discussion on how the authors would develop this study to other target groups like singles, newly wed couples of different socio-economic strata.

Reviewer 2 Report

Thank you for the opportunity to review this manuscript. I enjoyed reading it and recommend its publication with some small editorial changes.

I am attaching 1 file, which is the manuscript with highlights. Mostly yellow. These are spots that need editorial attention. The green highlight is a suggestion to move a sentence. The blue highlight is on the terms GIRLS and BOYS. I might suggest changing this to FEMALES and MALES, but will leave it up to the authors and journal. If you do change it make sure to change it in the tables too.

Here are my specific edits...

Line 41-41

Reword…

However, China’s fertility rate continued to decline, falling into what international academics classify as a very low fertility level by 2021.

Line 52-54

First, the term ‘mediating’ has a specific statistical interpretation which does not align with this sentence. Second, the correct term is ‘inter’pregnancy interval (IPI) not ‘intra’.

I would eliminate this sentence altogether and progress to next sentence, changing one word… “Investigating couples’ fertility intentions and related factors is essential to inform the fertility policy for its successful implementation.

Line 63

Delete “which may widen the gap between intentions and behaviors”

Line 65

Change “will recruit” to … we recruited

Line 71

Change “… can provide the basis for policies…” to can inform policies

Line 85

20-49

Line 105

Change “intra-pregnancy” interval to “inter-pregnancy interval (IPI)

Line 143

The authors state “calculated the median”, but do not give a rationale. When the median is the only measure of central tendency it suggests the data is skewed. There is no mention of this.

Line 166

How was the target number of 1,392 calculated?

Line 194

Change ‘think’ to thought

Line 210

Change “decreases’ to decreased

Line 212

Missing a word … “…more likely to have MORE children …” (or more likely to have another child)

Line 215

Change ‘have’ to had

Line 218

Eliminate the word ‘simultaneous’

Line 219-221

Need to reword…

This STUDY used a population-based survey TO INVESTIGATE fertility intention to have a third child AMONG MARRIED COUPLES IN CHINA WITH TWO CHILDREN. It ALSO examined the factors influencing intention TO ACCESS useful information FOR policymakers about the short-term effects of the NEW three-child policy.

Line 222-224

Revise sentence. Make it shorter. I suggest…

OVERALL, PARTICIPANTS HAD a low level of knowledge of the three-child policy and an extremely low intention of a third birth. Notably, THOSE FAMILIES with two girls were MORE willing to have one more child.

Line 230-232

Move this line forward …. See green notes

Line 256

Change “is interesting” to “would provide further insight”.

Line 283-285

Revise sentence to … “Many studies HAVE established parenthood as a protective factor against separation or divorce [32], WHICH IS IMPORTANT IN THE Chinese familism culture. IN CHINA, FAMILISM NOT ONLY PRIORITIZES FAMILY, BUT ALSO advocates the continuity of the family line as a life mission. IN OTHER WORDS, fertility benefits family stability.”

Finally, I inserted a comment under limitations. I think you should collect information about mode of delivery going forward if possible.

Again, I enjoyed the manuscript.

I hope my comments are helpful.

Reviewer 3 Report

October 20, 2022

Review of the manuscript entitled "Fertility intention to have a third child in China following the 2 three-child policy: A cross-sectional study" submitted to the International Journal of "Environmental Research and Public Health". The study examines intention of having a third child in an unrepresentative sample of population aged 20-49 in China. The results can be useful for the new policy pronatalist policy in China. However, the study has serious methodological and theoretical limitations that make it unsuitable for publication in this journal. I have highlighted some of these limitations below if the authors to choose to revise it.  

-          The results are clearly biased toward WeChat users in China, and not being representative of the national population. This limitation should be acknowledged and all generalizations and recommendations should be done with this consideration. So, the Discussion section includes generalizations to the national population based on this biased results.

-          LL102-122: There is a serious lack of a conceptual model, defining and explain the theoretical logic behind choosing covariates of fertility intention, especially a third child that is normally different from the intention of the first and second child.

-          L85: typo: revised age range to "20-49"

-          L115: the name of "Fertility intention scale" does not represent the content, as determinants of fertility intentions. Besides, the theoretical and conceptual base of selecting this composite measure has not been explained.

-          L143: what is IQR? Write the full name.

-          L166: How did authors define and access the target group of 1,392? This is a "convenient" sampling method. Why 1392?

-          Table 3: What is M(IQR)? Clearly explain, please what are these figures in Table 3.

-          Figure 3. Authors need to provide the original results of the full models of logistic regressions, instead of just a graph.
